# Uptake of Human Papillomavirus Vaccine and Intention to Vaccinate among Healthy Pregnant Women in Serbia: A Cross-Sectional Study on Awareness, Knowledge, and Attitudes

**DOI:** 10.3390/v13050727

**Published:** 2021-04-21

**Authors:** Ljiljana Markovic-Denic, Srboljub Milicevic, Jovana Todorovic, Vladimir Nikolic, Olivera Djuric, Vuk Marusic, Stefan Dugalic, Brankica Vasiljevic, Miroslava Gojnic-Dugalic

**Affiliations:** 1Faculty of Medicine, Institute of Epidemiology, University of Belgrade, Dr Subotica Starijeg 8, 11000 Belgrade, Serbia; vladimir.nikolic@gmail.com (V.N.); vuk.marusic@med.bg.ac.rs (V.M.); 2Clinic for Obstetrics and Gynecology, Clinical Centre of Serbia, Koste Todorovica 26, 11000 Belgrade, Serbia; srbazarube@gmail.com (S.M.); stef.dugalic@gmail.com (S.D.); miroslavagojnicdugalic@yahoo.com (M.G.-D.); 3Faculty of Medicine, Institute of Social Medicine, University of Belgrade, Dr Subotica Starijeg 15, 11000 Belgrade, Serbia; jovana.todorovic@med.bg.ac.rs; 4Epidemiology Unit, Azienda USL-IRCCS di Reggio Emilia, 42122 Reggio Emilia, Italy; olivera.djuric@ausl.re.it; 5Section of Public Health, Center for Environmental, Nutritional and Genetic Epidemiology (CREAGEN), Department of Biomedical, Metabolic and Neural Sciences, University of Modena and Reggio Emilia, Via Università, 4, 41121 Modena, Italy; 6Maternity and Child Health Service, NMC Royal Hospital DIP, Dubai Investments Park 1, 00000 Dubai, Saudi Arabia; brankica.vasiljevic1@gmail.com

**Keywords:** human papillomavirus, vaccine, pregnancy, attitudes, knowledge

## Abstract

We aimed to assess awareness, knowledge, and attitudes of healthy pregnant women towards human papillomavirus (HPV), to estimate factors associated with a positive attitude towards HPV immunization and to assess the uptake of the vaccine among their children. A cross-sectional study was conducted at the University Clinic of Gynecology and Obstetrics, Belgrade, Serbia among pregnant women attending their regular gynecological check-ups at the 12th gestational week. Knowledge about HPV and HPV vaccine was assessed using a specifically designed 12-item and 5-item questionnaires. Out of total 265 included women, 79.3% had heard of HPV, and 37.5% knew that HPV vaccine exists. HPV vaccine knowledge score was associated with higher odds for a positive attitude towards vaccination of both female (OR = 4.10, 95% CI 1.50–11.29) and male (OR = 3.71, 95% CI 1.52–9.01) child. The number of children (OR = 1.32, 95% CI 1.04–1.67) and high vaccine knowledge score (OR = 1.64 95% CI 1.13–2.39) were independent predictors associated with willingness to vaccinate child against HPV. The gynecologist was the preferable point of reference for information seeking about the HPV vaccine. Despite relatively high HPV awareness and knowledge among pregnant women in Serbia, about one-third of them are HPV vaccine aware, and are willing to vaccinate their children against HPV.

## 1. Introduction

Human papillomavirus (HPV) is considered to be the most common sexually transmitted infectious agent today [1]. The estimated worldwide prevalence of HPV infection in the general population is approximately 12% with significant regional differences [2]. Around 75% of sexually active people are infected with HPV at some point during their lifetime [2], however the majority of HPV infections will resolve on their own and will be asymptomatic [2]. However, some will be persistent and can lead to precancerous lesions and cancer [2]. The estimated prevalence of infection with HPV 16 and/or 18, that are associated with almost 70% of all cases of invasive cervical cancer, in Southern Europe, which Serbia belongs to, is around 3.8% [3].

Cervical cancer in Serbia is the fourth leading cancer among women and the second most common cancer among women aged 15–44 [3]. Many efforts have been made to decrease the incidence of cervical cancer, including organized cervical cancer screenings for all women aged 25 to 69, and the availability of vaccination for HPV [4]. Around three-quarters of all cases of cervical cancer, along with a significant proportion of all cases of anal, vulva, vaginal, penile, and cancer of oropharynx, are associated with an HPV infection with types 16 and 18, which are part of all HPV vaccines. The vaccination provides significant public health possibilities for primary prevention of the disease and the reduction of HPV prevalence is considered as one of the public health priorities globally [5,6,7].

Three types of HPV vaccine are available in Serbia, bivalent, quadrivalent, and nine-valent HPV vaccine [8]. The voluntary vaccination against HPV began in 2016, as a part of the pilot project ‘Improving the prevention of the diseases associated with HPV’. In 2017, the vaccine was introduced at the national level as a recommended vaccine, and to this day it is such according to the law [9].

According to the guidelines, it is recommended for all children aged 11 and 12, teenagers who have not been vaccinated, and young women up until the age of 26 and young men up until the age of 21 [10]. Yet, the HPV vaccine initiation coverage in Serbia is very low (2.0%) [11]. The main obstacle to the adequate implementation of vaccination against HPV is the lack of knowledge on HPV, on its epidemiological characteristics, modes of transmission, and association with malignant diseases which can have negative influence on effectiveness of vaccination programs [12]. The factor which was shown to be associated with the vaccination uptake among adolescents was also the maternal educational level, however, the effect of maternal education is not consistent across countries [13].

Except mother’s educational level and parental acceptance of the vaccine in general [11,13], the facilitators of vaccination against HPV reported so far are the physician’s recommendation, peer encouragement, and health insurance coverage [13]. The main barriers reported are the vaccine price, parental beliefs that their children are at low risk of contracting an HPV infection, and parental concerns regarding vaccine efficacy and safety [13,14,15]. The most common parental concerns are associated with belief that their children are too young to be sexually active or that the vaccination may encourage risky sexual behavior. The parents are also commonly worried that their knowledge on HPV and vaccination is insufficient [13].

The aim of this study was to examine the HPV and HPV vaccine-related knowledge among healthy pregnant women in Serbia, to assess factors associated with willingness to vaccinate their children, and to estimate the uptake of at least one dose of vaccine received by their children.

## 2. Materials and Methods

This was a cross-sectional study, conducted in February 2020 among healthy pregnant women who attended the regular prenatal check-ups for the screening of fetal aneuploidies and morphological anomalies (“double-test”) performed at the 12th–14th gestational week at the Clinic of Gynecology and Obstetrics, Clinical Centre of Serbia, Belgrade. This University Clinic is one of the three Gynecology and Obstetrics clinics in the capital of Serbia. They are available for all pregnant women because the mandatory social security insurance, which exists in our country, allows access to any of them. The double test screening is recommended for all pregnant women in Serbia and, in Belgrade, is being done in tertiary institutions only. A total of 256 consequent women who attended the check-up at the time of the study were included.

### 2.1. Study Instrument

Knowledge and attitudes about the HPV and HPV vaccine were assessed using a specifically designed questionnaire based on the available scientific literature and, also, the official brochure “Prevention of the diseases caused by Human Papillomavirus (HPV)–HPV Vaccine” published by the Institute of Public Health of Serbia, “Dr. Milan Jovanović-Batut” (http://www.batut.org.rs/download/aktuelno/rgm2019/Brosura%20HPV.pdf, accessed on 1 March 2021). We previously used this questionnaire in our study on women referred to colposcopy and/or HPV testing and on a sample of medical students [16].

The questionnaire consisted of three sections. The first regarded socio-demographic characteristics of the participants (age, birthplace, place of residence, education, marital status, employment status). The second part of the questionnaire referred to the gynecological and obstetrics patient history (age at menarche, age at the first sexual intercourse, number of sexual partners, number of sexual partners in the past three months, number of pregnancies, and deliveries, date of the last menstruation). The third section regarded the questions on HPV awareness and knowledge, and HPV vaccine awareness, along with attitudes towards the HPV vaccination.

HPV awareness and HPV vaccine awareness were assessed with a Yes/No question: ‘Have you ever heard of the human papillomavirus?’ and ‘Is there a vaccine available against human papillomaviruses?’ All participants who answered ‘Yes’ to these questions were classified as HPV aware and HPV vaccine aware, respectively. The knowledge on HPV was assessed with 12 questions regarding the epidemiological and clinical characteristics of HPV. The knowledge on HPV vaccine was assessed with 5 questions about its availability in Serbia, recommended use, and its association with cervical cancer. Possible responses to all these questions were ‘yes’, ‘no’, or ‘I don’t know’. Each correct answer was scored as one, and the total score on these scales could vary between 0 and 12 for HPV knowledge score and between 0 and 5 for HPV vaccine knowledge score. HPV and HPV vaccine knowledge scores were analyzed only among women ascertained as HPV or HPV vaccine aware. Ascertainment of HPV awareness (confirmation that woman who answered “yes” to the single question on HPV awareness really knew what HPV was) was done in the data analysis, i.e., if the knowledge score of the women was 0, that woman was not considered HPV aware regardless of the fact that she responded “yes” to the question “Have you ever heard of HPV?” These women were therefore reclassified to the “HPV not aware” group in the further analysis.

Both HPV knowledge scales, one about HPV and the other about HPV vaccine, showed good internal consistency in our sample (Cronbach’s α = 0.896 and 0.803, respectively).

### 2.2. Statistical Analysis

Descriptive and analytical statistical methods were used in the data processing. The internal consistency and reliability of the HPV and HPV vaccine knowledge scale were assessed by calculating the Cronbach’s alpha coefficient. The normality of the distribution of continuous variables was evaluated by using visual inspection of histograms and probability plots. Data were presented as mean ± SD or median (interquartile range [IQR]) for continuous variables, depending on the normality of data distribution, and number (percentage) for categorical variables. Differences in knowledge scores between the categories of patients’ characteristics were assessed using the Mann–Whitney U test. A Chi-square or Fisher’s exact test was used for the analysis of categorical data, where appropriate. Univariate logistic regression analyses were performed for HPV awareness, HPV vaccine awareness, HPV vaccine knowledge, and willingness to vaccinate themselves as an outcome. The multivariable logistic regression model was performed to identify factors associated with the willingness to vaccinate their child against HPV. Variables that were significantly associated with these outcomes at the significance level <0.1 in the univariate logistic regression analysis and based on clinical significance were entered into the multivariable logistic regression model. Odds ratios (OR) with 95% confidence intervals (CI) were computed and the Hosmer–Lemeshow goodness-of-fit test was performed to assess overall model fit. The statistical analysis was performed using SPSS version 23.0 software (SPSS Inc., Chicago, IL, USA).

## 3. Results

### 3.1. Demographic Characteristics and Gynecological Data

The average age of the pregnant women included in the study was 33 ± 7.1 years (Table 1). Slightly more than half of the women (55.5%) had a university degree (>12 years of schooling), followed by high school (9–12 years of schooling) (38%), while 6.3% of women completed elementary school (eight years of schooling). The largest number of women lived in a marital or extramarital union with a partner (96.9%) in an urban environment (87.8%) and were employed (74.2%).

### 3.2. HPV and HPV Vaccine Awareness

Out of a total of 256 women included in the study, 203 (79.3%) had heard of HPV while substantially fewer women, 96 of them (37.5%), knew that the HPV vaccine exists. 

Odds ratios with 95% confidence intervals along with median knowledge scores about HPV and HPV vaccine are presented in Table 2. Secondary school education and university degree were strongly associated with HPV awareness (OR 6.34 95% CI 2.00–20.03, OR 17.19 95% CI 5.29–55.84, respectively). Women who live in urban areas (OR = 2.52, 95% CI 1.12–5.66) and are employed (OR = 2.40, 95% CI 1.26–4.59) were more likely to hear of HPV than women who live in rural areas and are not employed.

Factors associated with the highest odds of having heard of HPV vaccine were older age (OR = 1.07, 95% CI 1.03–1.12), university degree compared to high school graduation (OR = 4.55, 95% CI 2.50–8.27), urban compared to rural residence (OR = 3.58, 95% CI 1.42–10.40), while having had one or more abortions was associated with a lower chance of having heard of HPV vaccine (OR = 0.08, 95% CI 0.01–0.70).

### 3.3. HPV and HPV Vaccine Knowledge

Twelve questions used to assess HPV knowledge are presented in Table 3 along with five questions used to assess HPV vaccine knowledge. The majority (86.2%) of HPV aware women knew that HPV can be transmitted by sexual contact, that the risk of infection increases with the number of sexual partners (72.4%), that HPV can cause genital warts (71.9%), cervical carcinoma (68.2%), and infection in men (67.5%), and that there was more than one type of HPV (59.1%). About half of them (50.2%) knew that HPV could be transmitted vertically, and that infection could be asymptomatic (52.7%). Less than half of women (33.0%) knew that HPV can be transmitted through towels, toilets, swimming pools, etc., that HPV infection can be cured spontaneously (13.8%), and that most sexually active people get HPV at some time in their lifetime (45.8%). The minority (9.9%) knew that condoms cannot prevent the occurrence of infection caused by HPV. The median knowledge score about HPV was 7.0 (IQR = 5.0).

About two-thirds (64.6%) of vaccine-aware women knew that the vaccine was available in Serbia. Most of them knew (70.8%) that the vaccine was recommended before the first sexual intercourse and that vaccinated women should continue undergoing cervical cancer screening even after vaccination (72.9%). About half of women (56.3%) knew that vaccines cannot cause cervical cancer. One-third (32.3%) of respondents knew that the vaccine was also recommended for men. The median knowledge score about HPV was 3.0 (IQR = 2).

### 3.4. Sources of Information

The majority of women used Internet sites of official organizations and this source of knowledge was significantly more often reported by women who had a score of HPV knowledge greater than seven while women who had a score of HPV knowledge ≤7 used other sources of information in larger numbers (Figure 1). Gynecologists (87.1%) are the first option for seeking an opinion on the vaccine as it is shown in Figure 2.

### 3.5. HPV Vaccine Coverage

Only 7 out of 232 women (3%) responded that their child had been already vaccinated against HPV with at least one dose of the vaccine, without a significant difference between low and high HPV knowledge groups (2.1% vs. 5.3%, *p* = 0.120).

### 3.6. Attitudes towards Vaccination

Willingness to vaccinate themselves or their children against HPV increased substantially with increased awareness that vaccine exist and with a higher score of knowledge about HPV vaccine (Table 4). HPV vaccine knowledge score greater than 3 was highly associated with higher odds for a positive attitude towards vaccination of both female (OR = 4.10, 95% CI 1.50–11.29) and male (OR = 3.71, 95% CI 1.52–9.01) child.

Multivariate logistic regression with OR and 95% CI for factors associated with willingness to vaccinate child against HPV among pregnant women is presented in Table 5. Among five variables associated with willingness to vaccinate her child in univariate analysis, number of children (OR = 1.32, 95% CI 1.04–1.67) and vaccine knowledge score (OR = 1.64 95% CI 1.13–2.39) remained independently associated with the outcome in the multivariate model. Similar results were obtained when subanalysis stratified by sex of the future child was performed (data not shown).

## 4. Discussion

The present study examined the awareness and knowledge of healthy pregnant women in their first trimester of pregnancy in Serbia on HPV, along with their attitudes on vaccination, factors associated with willingness to vaccinate their children, and actual uptake of the HPV vaccine.

Our study showed that almost 80% of our participants stated that they have heard about HPV before, but much less, only just over a third was aware that there is an HPV vaccine. In our previous study, conducted seven years before the current, among women with positive PAP smear test obtained at the primary care facilities and referred to the hospital for further examination, it was found that 60.5% of women had heard of HPV [14]. The interest in HPV has certainly grown between two of our studies, and much more information about this virus is available today. Besides, in the current study, the study population consisted of pregnant women who could receive a lot of information in counseling centers for pregnant women that function in every health center in our country. Public education campaigns regarding cervical cancer screening, which are organized over several years in our country [17], probably had an impact on raising the level of awareness and HPV knowledge, which is confirmed in other studies [18]. In the first study conducted in our country among parents presented in two primary health care centers in Belgrade, because of any health problem with their child aged ≤18 years, a knowledge gap on HPV was observed. However, this population of parents was older than our participants. Our results are identical to the results from the study conducted almost a decade ago in the United States of America [19]. A study recently conducted among young pregnant adolescents in Brazil [12] suggested that about 80% of them were aware of HPV, which is similar to our results. Besides high HPV awareness, participants showed a high knowledge level on HPV infection in our study. The median knowledge score among HPV-aware women was 7 out of 12. The highest proportion of women (more than 86%) knew that HPV can be transmitted by sexual contact. This is higher than in the study in neighboring Romania among women in the general population that showed that just under 65% of women know that HPV can be transmitted through sexual contact, with both studies having a similar mean age of the participants [6]. Only a small proportion of participants knew that condom cannot prevent the transmission of infection. However, they knew that risk of infection increase with several sexual partners and that, besides cervical cancer, HPV can cause genital warts, which is comparable with results of other studies [20]. In our study, only one-quarter of the sample knew that HPV cannot be transmitted through towels, toilets, or swimming pools and almost only one in eight participants knew that HPV infections can be cured spontaneously.

HPV awareness was significantly associated with educational level in our study, as women with secondary education had more than 6.5 times higher likelihood to have heard about HPV and those with tertiary education had 17.2 times higher likelihood to be aware of HPV compared to participants with primary education only. Serbia has a well-developed network of primary health care centers with gynecological health care services and routine pregnancy counseling sessions for pregnant women. These sessions could include the provision of information on HPV with a special focus on the provision of proper, detailed, and tailored information such that even women with lower education could benefit from.

It has been more than ten years since the HPV vaccine was introduced in EU countries [21]. In Serbia, it has only been a few years since this vaccine was recommended according to law regulations [9], although vaccination is still not widespread. A negative effect of HPV infections on pregnancy outcome like preterm birth and premature rupture of membranes is well known [22]. Knowledge about these findings could motivate pregnant women for vaccination. Only a quarter of our respondents expressed their willingness to be vaccinated, but more than half of those had better knowledge about the HPV vaccine. Additionally, significantly higher proportions of HPV vaccine-aware women, with a score of knowledge higher than the median score, were found to believe that they would vaccinate their female (39.5%) as their male child (30.1%). This is consistent with previous findings on HPV vaccination intentions of women in Italy [23] and the United States [19]. The sex of the child has been recently confirmed in a meta-analysis to have a moderation effect on actual vaccination uptake which is twofold higher in girls due to the higher vaccine offering to them. i.e., to those who are believed will benefit from the most [24]. The lack of knowledge on the importance of vaccinating both male and female adolescents can have significant negative effects on public health efforts to increase the vaccination uptake among adolescents and to decrease the prevalence of HPV infection in the general population. Another factor that influences the willingness to vaccinate may be the costs associated with the vaccine, as, although the HPV vaccination is recommended in Serbia, the vaccine is not covered by the health insurance and the three doses are paid out-of-pocket. This may have a significant influence, especially among the women of lower socio-economic status. The public health efforts in different settings may include the efforts for the increase of the availability of the HPV vaccines to the general population.

The trust in the vaccine safety is considered one of the main factors influencing the vaccination uptake. Unfortunately, recent analysis showed the decrease in the belief that vaccines are safe in Serbia and that general trend may have influenced the willingness for uptake of HPV vaccine as well. This presents the rising global public health issue and calls for joint public health action [25].

Women with a higher number of children had 1.32 times higher likelihood to be willing to vaccinate a child against HPV, while with each point increase in the vaccine knowledge score, the likelihood for willingness to vaccinate increased by 64%. These results also strengthen the importance of adequate educational programs and as multiple previous studies have shown [7,26] higher knowledge on the vaccine can significantly improve the likelihood for vaccination.

Almost 90% of the participants reported that they would be seeking information on HPV vaccination from their gynecologist. This is a good indicator of the trust between pregnant women and their chosen gynecologists, which was also shown previously, as frequent prenatal visits improve the relationship between patients and physicians [26].

Given the fundamental role of parents in HPV vaccine uptake of their children, public health strategies should address modifiable factors that influence parents’ uptake, in particular increasing women’s awareness of the vaccination benefits and addressing their fears about vaccine safety. Particular clinical importance reflects in the role of gynecologists in offering the vaccine and communicating its benefits to pregnant women for their female and male children. Motivation and education of the clinicians to promote and offer the HPV vaccine to the pregnant women together with other public health strategies to educate women on this topic could have an important role in accelerating the vaccine uptake.

The main limitation of this study is that it was conducted in only one university obstetric clinic. However, all three obstetric clinics in Belgrade are close to each other and are all accessible to women due to public social security. Moreover, the study was conducted at the largest clinic for gynecology and obstetrics in the capital of Serbia. Besides, the double test is mandatory for all pregnant women at the end of the first trimester of pregnancy and it is available only at university clinics in our capital, and not in primary health care centers, which are the most common centers for other examinations during pregnancy. Another possible limitation is that this cross-sectional study was conducted during the first trimester. Expectant mothers may think more about the vaccination of their child before the end of pregnancy and ask for more information about all vaccines, which would increase their level of knowledge about the HPV vaccine. However, it was found that the prevalence of HPV is higher during pregnancy, due to hormonal changes in the body of expectant mothers. The viral replication is very high during the second trimester when the highest prevalence of HPV was observed [27]. For that reason, we believe that it is important to assess the level of knowledge of pregnant women in early pregnancy, and to encourage education about the HPV virus and the HPV vaccine during pregnancy through various training programs within the counseling center for pregnant women. One of the limitations is the cross-sectional design of our study which did not allow the final temporal and causal inference. Following up with the pregnant women until the end of pregnancy would enable the observation of changes in the level of their knowledge and changes in attitudes about their own or vaccination of their children. There is a need to perform such a study. However, this is one of only a few studies examining the uptake of the HPV vaccine, level of knowledge about the HPV vaccine, and willingness of healthy pregnant women to vaccinate their child and the results could be used for the creation of tailored educational programs aiming to increase the level of knowledge on HPV and HPV vaccine among future mothers. However, self-reported survey methodology could be a source of reporting bias. The highly selected population of pregnant women attending their regular check-ups surely limits the external validity of the results which might could not be representative of the awareness and knowledge of the non-pregnant women of the same age (overestimate) or the women who already have children of the age in which the HPV vaccine is suggested (underestimate). The second is supported by our finding that women with more children have higher odds of vaccinating their child.

## 5. Conclusions

Our study showed that about 80% of pregnant women in Serbia are HPV aware, but almost two-thirds of them are not HPV vaccine aware. Concerningly, only one-third of them are willing to vaccinate their children against HPV. As vaccine knowledge level is the independent predictor of vaccination willingness of currently pregnant women who are soon to be mothers of youth for whom the HPV vaccine is intended, there is a necessity for the introduction of the educational programs that would include wide populations of current and future mothers.

## Figures and Tables

**Figure 1 viruses-13-00727-f001:**
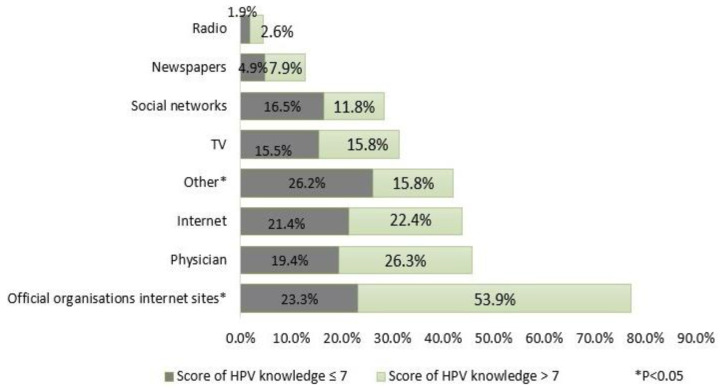
Sources of information about HPV.

**Figure 2 viruses-13-00727-f002:**
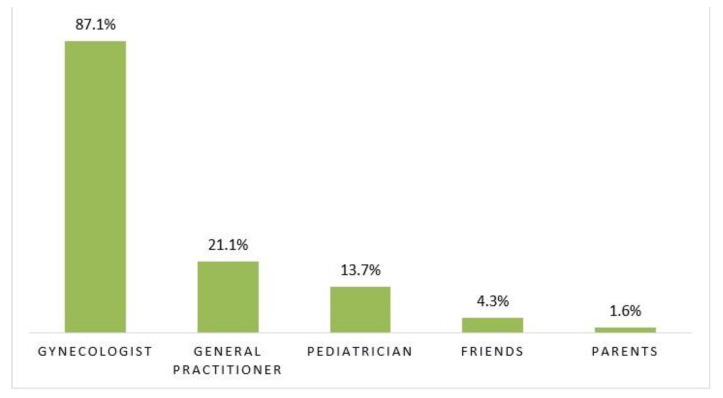
Preferable sources of information about HPV vaccine.

**Table 1 viruses-13-00727-t001:** Demographic characteristics and gynecological data.

	Total (*n* = 256)
**Demographic characteristics**	***n* (%)**
Age **	33.0 ± 7.1
Level of education—school	
Elementary	16 (6.3)
Secondary	97 (38.2)
Faculty	141 (55.5)
Marital status	
Married/Cohabiting	245 (96.9)
Single	8 (3.2)
Residence	
Rural	31 (12.2)
Urban	223 (87.8)
Employment	
Non-employed	65 (25.8)
Employed	187 (74.2)
Gynecological data	
Age at menarche (years) **	12.9 ± 1.5
Age at first sexual intercourse (years) **	19.1 ± 3.1
Treated for infertility	27 (10.5)
Number of abortions	
0	179 (69.9)
≥1	68 (26.6)
Missing	9 (3.5)
Number of children	
0	114 (44.5)
≥1	119 (46.5)
Missing	23 (9.0)
Number of sexual partners (total)	
1	65 (25.4)
2–5	120 (46.9)
≥6	31 (12.1)
Missing	40 (15.6)
Number of sexual partners (in last 3 months)	
0	15 (5.9)
1	206 (80.5)
2–5	0 (0.0)
≥6	0 (0.0)
Missing	35 (13.7)

* Median (IQR); ** Mean ± SD.

**Table 2 viruses-13-00727-t002:** Odds ratios and 95% CI for HPV and HPV vaccine awareness along with corresponding knowledge scores.

	Ever Heard of HPV (n = 203)			Ever Heard of HPV Vaccine (n = 96)		
	n (%)	OR (95% CI)	Knowledge score *	n (%)	OR (95% CI)	Knowledge score *
**Demographic characteristics**						
**Age ****	33.4 ± 7.2	1.05 (1.00–1.10)	/	35.1 ± 7.3	**1.07 (1.03–1.12)**	/
**Level of ducation**						
Elementary	5 (2.5)	ref.	2 (4)	/	/	/
Secondary	72 (35.6)	**6.34 (2.00–20.03)**	**5 (5)**	20 (20.8)	ref.	3 (1)
Faculty	125 (61.9)	**17.19 (5.29–55.84)**	**7 (4)**	76 (79.2)	**4.55 (2.50–8.27)**	3 (2)
**Marital status**						
Married/Cohabiting	197 (97.5)	ref.	7 (4)	93 (97.9)	ref.	3 (2)
Single	5 (2.5)	0.41 (0.09–1.76)	8 (9)	2 (2.1)	0.62 (0.12–3.28)	2 (0)
**Residence**						
Rural	20 (9.9)	ref.	4 (6)	5 (5.2)	ref.	3 (2)
Urban	183 (90.1)	**2.52 (1.12–5.66)**	7 (4)	91 (94.8)	**3.85 (1.42–10.40)**	3 (2)
**Employment**						
Non-employed	44 (22.0)	ref.	6 (5)	18 (18.8)	ref.	3.5 (2)
Employed	156 (78.0)	**2.40 (1.26–4.59)**	7 (4)	78 (81.3)	1.85 (0.99–3.45)	3 (2)
**Gynecological data**						
**Age at menarche (years) ****	12.8 ± 1.5	**0.79 (0.65–0.97)**	/	12.8 ± 1.4	0.88 (0.73–1.05)	/
**Age at first sexual intercourse (years) ****	19.1 ± 3.1	1.01 (0.91–1.12)	/	19.7 ± 3.5	1.10 (1.01–1.20)	/
**Treated for infertility**	21 (10.3)	0.91 (0.35–2.39)	7 (5)	11 (11.5)	1.17 (0.51–2.67)	3 (2)
**Number of abortions**						
0	142 (70.0)	ref.	7 (5)	64 (66.7)	ref.	3 (2)
≥1	54 (26.6)	1.10 (0.22–5.50)	7 (5)	25 (26.0)	**0.08 (0.01–0.70)**	3 (2)
Missing	7 (3.4)	1.10 (0.21–5.90)	6 (4)	7 (7.3)	**0.08 (0.01–0.73)**	2 (1)
**Number of children**						
0	88 (43.3)	ref.	7 (5)	40 (41.7)	ref.	3 (2)
≥1	96 (47.3)	0.71 (0.22–2.28)	6 (4)	45 (46.9)	0.63 (0.26–1.56)	3 (1.5)
Missing	19 (9.4)	0.88 (0.27–2.83)	7 (3)	11 (11.5)	0.70 (0.28–1.72)	2 (1)
**Number of sexual partners (total)**						
1	50 (24.6)	ref.	6 (4)	25 (26.0)	ref.	3 (1)
2–5	92 (45.3)	0.48 (0.16–1.43)	7 (5)	44 (45.8)	1.32 (0.57–3.03)	3 (2)
≥6	26 (12.8)	0.47 (0.17–1.31)	7.5 (4)	14 (14.6)	1.24 (0.58–2.66)	4 (2.3)
Missing	35 (17.2)	0.74 (0.19–2.83)	6 (4)	13 (13.5)	1.75 (0.66–4.66)	2 (1.5)
**Number of sexual partners (in last 3 months)**						
0	10 (4.9)	ref.	6 (4)	5 (5.2)	ref.	5 (3.5)
1	165 (81.3)	0.50 (0.13–1.94)	7 (5)	76 (79.2)	0.70 (0.19–2.55)	3 (2)
2–5	0 (0.0)	/	/	0 (0.0)	/	/
≥6	0 (0.0)	/	/	0 (0.0)	/	/
Missing	28 (13.8)	1.01 (0.41–2.46)	7 (3)	15 (15.6)	0.78 (0.37–1.63)	2 (1)

CI, Confidence interval; HPV-human papillomavirus. Values in bold are statistically significant. * Median (IQR); ** Mean ± SD.

**Table 3 viruses-13-00727-t003:** HPV and HPV vaccine knowledge questionnaires.

HPV Knowledge Item (Correct Answer)	Yes	No	I Don’t Know	Total Number of Answers
Total Women Included 203	*n* (%)	*n* (%)	*n* (%)	*n* (%)
Is there more than one type of human papilloma virus? (yes)	120 (59.1)	5 (2.5)	75 (36.9)	200 (98.5)
Can HPV be transmitted by sexual contact? (yes)	175 (86.2)	3 (1.5)	22 (10.8)	200 (98.5)
Can human papillomaviruses be transmitted through towels, toilets, swimming pools, etc.? (no)	67 (33.0)	54 (26.6)	81 (39.9)	202 (99.5)
If a pregnant woman is infected with human papilloma viruses, can they be transmitted to the newborn during childbirth? (yes)	102 (50.2)	13 (6.4)	85 (41.9)	200 (98.5)
Can person infected by HPV have no symptoms? (yes)	107 (52.7)	24 (11.8)	69 (34.0)	200 (98.5)
Does the risk of human papillomavirus infection increase with the number of sexual partners? (yes)	147 (72.4)	11 (5.4)	42 (20.7)	200 (98.5)
Can HPV cause infection in men? (yes)	137 (67.5)	10 (4.9)	54 (26.6)	201 (99.0)
Can infections caused by HPV be cured spontaneously? (yes)	28 (13.8)	121 (59.6)	50 (24.6)	199 (98.0)
Can HPV cause genital warts? (yes)	146 (71.9)	4 (2.0)	52 (25.6)	202 (99.5)
Can HPV cause cervical carcinoma? (yes)	131 (68.2)	4 (2.1)	57 (29.7)	
Do most sexually active people get human papillomaviruses at some time in their lifetime? (yes)	93 (45.8)	27 (13.3)	81 (39.9)	201 (99.0)
Can condoms prevent occurrence of infection caused by HPV? (no)	158 (77.8)	20 (9.9)	24 (11.8)	202 (99.5)
Median (IQR) knowledge score	7.0 (5.0)			
HPV vaccine knowledge item (correct answer)				
Total women included 96				
Is the vaccine available in Serbia? (yes)	62 (64.6)	5 (5.2)	29 (30.2)	96 (100.0)
Is the vaccine recommended before first sexual intercourse? (yes)	68 (70.8)	1 (1.0)	27 (28.1)	96 (100.0)
Whether vaccinated women should go for cervical cancer screening? (yes)	70 (72.9)	0 (0.0)	25 (26.0)	95 (99.0)
Whether the vaccine can cause cervical cancer? (no)	14 (14.6)	54 (56.3)	28 (29.2)	96 (100.0)
Is the vaccine also recommended for men? (yes)	31 (32.3)	16 (16.7)	49 (51.0)	96 (100.0)
Median (IQR) knowledge score	3.0 (2.0)			

HPV–human papillomavirus.

**Table 4 viruses-13-00727-t004:** Attitudes towards vaccination.

Attitudes towards Vaccination	Total (*n* = 237)	Total Vaccine Aware (*n* = 96)	HPV Vaccine Knowledge Score ≤ 3 (*n* = 61)	Score of HPV Vaccine Knowledge > 3 (*n* = 35)	
	*n* (%)	*n* (%)	*n* (%)	*n* (%)	OR (95% CI)
Would be vaccinated	66 (25.8)	43 (44.8)	23 (38.3)	20 (57.1)	2.14 (0.92–5.01)
Would vaccinate her female child	101 (39.5)	62 (64.6)	33 (54.1)	29 (82.9)	**4.10 (1.50–11.29)**
Would vaccinate her male child	77 (30.1)	45 (46.9)	22 (36.1)	23 (67.6)	**3.71 (1.52–9.01)**
Will she vaccinate her child against diseases for which vaccination is mandatory	204 (79.7)	84 (87.5)	51 (87.9)	33 (97.1)	4.53 (0.53–38.52)

Values in bold are statistically significant.

**Table 5 viruses-13-00727-t005:** Factors associated with willingness to vaccinate child against HPV among pregnant women.

Variable	Multivariate Logistic Regression
	OR (95% CI)
Age	1.05 (0.98–1.13)
High education (>12 years of schooling)	2.47 (0.70–8.65)
Employed vs. not employed	0.68 (0.17–2.66)
Number of children	**1.32 (1.04–1.67)**
HPV vaccine knowledge score	**1.64 (1.13–2.39)**

HPV—human papillomavirus. Values in bold are statistically significant.

## Data Availability

Data will be made available upon request.

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
