# Peer review of "Uptake of Human Papillomavirus Vaccine and Intention to Vaccinate among Healthy Pregnant Women in Serbia: A Cross-Sectional Study on Awareness, Knowledge, and Attitudes"

_viruses, 2021, doi:10.3390/v13050727_

Round 1
Reviewer 1 Report
The study is interesting and rather well-constructed, although there are many studies on the subject now.
Here are my comments for the improvement of the article :
Were the questions tested on a sample of subjects to assess the comprehension of the questions?
There is little information on who the people in the study are and how they are recruited.
I think table 1 needs to be cut into two different tables :
The first table could present just the description of the sample and an additional table could present the ORs and knowledge scores.
The interest of the study is limited by its cross sectional design which does not allow for causality analysis and by this type of self-reported questionnaire which has a lot of reporting bias.
Author Response
Response to Reviewer 1 Comments
Dear Madame/Sir,
Thank you for your time and assistance. We appreciate very much your professional advice and accepted comments and suggestions in the text.
We accepted all suggestions in Abstract, Introduction, Methods, Results, and Discussion section.
Also, we added the recommended references.
Comments to the authors:
Point 1: Were the questions tested on a sample of subjects to assess the comprehension of the questions?
Response 1: Thank you for your comment. The questions regarding knowledge, attitudes and practices regarding the HPV were previously used in our study on women referred to colposcopy and/ or HPV testing and on a sample of Medical students.
We added this explanation.
Point 2: There is little information on who the people in the study are and how they are recruited.
Response 2: Thank you for your comment. We corrected the sample description.
Point 3: I think table 1 needs to be cut into two different tables:
The first table could present just the description of the sample and an additional table could present the ORs and knowledge scores.
Response 3: Thank you for your comment. We presented the results in two tables. (pages 6,7, and 8)
Point 4: The interest of the study is limited by its cross sectional design which does not allow for causality analysis and by this type of self-reported questionnaire which has a lot of reporting bias.
Response 4: Thank you for your comment. We agree that a cross sectional design has its limitations, as we discussed in the limitations section of our article.
‘One of the limitations is the cross-sectional design of our study which did not allow the final causality conclusion. Following up the pregnant women until the end of pregnancy would enable the observation of changes in the level of their knowledge and changes in attitudes about their own or vaccination of their children. There is a need to perform such a study. However, this is one of only a few studies examining the uptake of the HPV vaccine, level of knowledge about the HPV vaccine, and willingness of healthy pregnant women to vaccinate their child and the results could be used for the creation of tailored educational programs aiming to increase the level of knowledge on HPV and HPV vaccine among future mothers.’ (page 12)
Reviewer 2 Report
The presented manuscript analyzes the attitude toward HPV vaccination of pregnant women in a referral academic Institution. It should be acknowledged that the presented paper is certainly interesting, and focused on a very relevant point. In particular, Introduction section appears clear, and methods and results adequately presented. However, some specific concerns need to be addressed:
-The authors data are deeply influenced by the local regulatory, and medical resources available. More efforts should be done to clarify in the Discussion section how the presented data can be used in the overall worldwide scenario.
-Which is the main clinical message of the paper? Please, improve this point providing a wider discussion on this specific point
-Do you think that pregnant status inlfluenced the final results? Please cite and discuss this point.
Author Response
Response to Reviewer 2 Comments
Dear Madame/Sir,
Thank you for your time and assistance. We appreciate very much your professional advice and accepted comments and suggestions in the text.
We accepted all suggestions in Abstract, Introduction, Methods, Results, and Discussion section.
Also, we added the recommended references.
Comments to the authors:
The presented manuscript analyzes the attitude toward HPV vaccination of pregnant women in a referral academic Institution. It should be acknowledged that the presented paper is certainly interesting, and focused on a very relevant point. In particular, Introduction section appears clear, and methods and results adequately presented. However, some specific concerns need to be addressed:
Point 1: The authors data are deeply influenced by the local regulatory, and medical resources available. More efforts should be done to clarify in the Discussion section how the presented data can be used in the overall worldwide scenario.
Response 1: Thank you for your comment. We added the paragraph in the discussion section.
‘Another factor that influences the willingness to vaccinate may be the costs associated with the vaccine, as, although the HPV vaccination is recommended in Serbia, the vaccine is not covered by the health insurance and the three doses are paid out-of-pocket. This may have a significant influence, especially among the women of lower socio-economical status. The public health efforts in different settings may include the efforts for the increase of the availability of the HPV vaccines to the general population.
The trust in the vaccine safety is considered one of the main factors influencing the vaccination uptake. Unfortunately, recent analysis showed the decrease in the belief that vaccines are safe in Serbia and that general trend may have influenced the willingness for uptake of HPV vaccine as well. This presents the rising global public health issue, and calls for joint public health action. ‘ (page 11)
Besides, we added new reference 25:
Point 2: Which is the main clinical message of the paper? Please, improve this point providing a wider discussion on this specific point
Response 2: We have added a paragraph which addresses the overall and clinical importance of the study results.
‘Given the fundamental role of parents in HPV vaccine uptake of their children, public health strategies should address modifiable factors that influence parents’ uptake, in particular increasing women’s awareness of the vaccination benefits and addressing their fears about vaccine safety. Particular clinical importance reflects in the role of gynecologists in offering the vaccine and communicating its benefits to pregnant women for their female and male children. Motivation and education of the clinicians to promote and offer the HPV vaccine to the pregnant women together with other public health strategies to educate women on this topic could have an important role in accelerating the vaccine uptake.’ (page 12).
Also, we added that ‘self-reported survey methodology could be a source of reporting bias. ’
Point 3: Do you think that pregnant status inlfluenced the final results? Please cite and discuss this point.
Response 3: Thank you to the reviewer for pointing this out. To make clearer about the influence of pregnancy on the final results, we added the following paragraph:
‘The highly selected population of pregnant women attending their regular check-ups surly limits the external validity of the results which might could not be representative of the awareness and knowledge of the non-pregnant women of the same age (overestimate) or the women who already have children of the age in which the HPV vaccine is suggested (underestimate). The second is supported by our finding that women with more children have higher odds of vaccinating their child. ’ (pages 12-13).
Round 2
Reviewer 1 Report
Thanks to the authors for their new manuscript updated and correction, I think it is publishable in this current form